# Analysis of Comparative Transcriptome and Positively Selected Genes Reveal Adaptive Evolution in Leaf-Less and Root-Less Whisk Ferns

**DOI:** 10.3390/plants11091198

**Published:** 2022-04-28

**Authors:** Zengqiang Xia, Li Liu, Zuoying Wei, Faguo Wang, Hui Shen, Yuehong Yan

**Affiliations:** 1Key Laboratory of National Forestry and Grassland Administration for Orchid Conservation and Utilization, The National Orchid Conservation Center of China and The Orchid Conservation and Research Center of Shenzhen, Shenzhen 518114, China; xiazengqiang@cemps.ac.cn (Z.X.); wzy787117923@163.com (Z.W.); 2Shanghai Key Laboratory of Plant Functional Genomics and Resources, Shanghai Chenshan Botanical Garden, Shanghai 201602, China; 3CAS Center for Excellence in Molecular Plant Sciences, Shanghai Institute of Plant Physiology and Ecology, Chinese Academy of Sciences, 300 Fenglin Road, Shanghai 200032, China; liliu@cemps.ac.cn; 4University of Chinese Academy of Sciences, Beijing 100049, China; 5College of Life and Sciences, Shanghai Normal University, Shanghai 201602, China; 6Key Laboratory of Plant Resources Conservation and Sustainable Utilization, South China Botanical Garden, Chinese Academy of Sciences, Guangzhou 510650, China; wangfg@scib.ac.cn; 7Guangdong Provincial Key Laboratory of Applied Botany, South China Botanical Garden, Chinese Academy of Sciences, Guangzhou 510650, China

**Keywords:** whisk ferns, transcriptome, positive selection, morphological evolution, root, leaf

## Abstract

While roots and leaves have evolved independently in lycophytes, ferns and seed plants, there is still confusion regarding the morphological evolution of ferns, especially in whisk ferns, which lack true leaves and roots and instead only exhibit leaf-like appendages and absorptive rhizoids. In this study, analyses of comparative transcriptomics on positively selected genes were performed to provide insights into the adaptive evolution of whisk fern morphologies. Significantly clustered gene families specific to whisk ferns were mainly enriched in Gene Ontology (GO) terms “binding proteins” and “transmembrane transporter activity”, and positive selection was detected in genes involved in transmembrane transporter activities and stress response (e.g., sodium/hydrogen exchanger and heat shock proteins), which could be related to the adaptive evolution of tolerance to epiphytic environments. The analysis of TF/TR gene family sizes indicated that some rapidly evolving gene families (e.g., the GRF and the MADS-MIKC families) related to the development of morphological organs were commonly reduced in whisk ferns and ophioglossoid ferns. Furthermore, the WUS homeobox-containing (WOX) gene family and the knotted1-like homeobox (KNOX) gene family, both associated with root and leaf development, were phylogenetically conserved in whisk ferns and ophioglossoid ferns. In general, our results suggested that adaptive evolution to epiphytic environments might have occurred in whisk ferns. We propose that the simplified and reduced leaf and root system in whisk ferns is the result of reduction from the common ancestor of whisk ferns and ophioglossoid ferns, rather than an independent origin.

## 1. Introduction

Leaves, roots and stems are the three major vegetative organs of vascular plants, mainly functional for photosynthesis, absorption and anchoring, respectively [1]. Leaves and roots evolved multiple times over plant evolutionary history [2,3,4,5,6,7,8,9]. Fossil evidence and studies on developmental pathways indicated several independent origins of tracheophytes [4,5,6]. Roots had evolved firstly in lycophytes in a bifurcating form in the Early Devonian [5,7,9]. The second origin event occurred independently in the ancestor of euphyllophytes in the Middle Devonian, which resulted in the genesis of adventitious roots and lateral roots in the monilophytes [5,7,9]. The third root-origin event occurred in the seed plants, resulting in primary roots in addition to adventitious and lateral roots [5,10]. On the other hand, based on fossil evidence and the distinct developmental mechanisms of extant vascular plants, it is now accepted that leaves originated in lycophytes and euphyllophytes independently, producing microphylls and megaphylls, respectively [2,3,9]. In addition, the comparison between fossils of the most ancient ferns and modern seed plants suggested that the clades following parallel traces of evolution may also be of independent origins [8].

Although the key functional organs such as roots and leaves have definitely evolved in Paleozoic vascular plants, their structures can be highly variable, as a result of special functional requirements and adaption to specific environments. Psilotaceae (whisk ferns), including *Psilotum* and *Tmesipteris*, do not produce leaves and roots, and only exhibit leaf-like appendages and absorptive rhizoids [11,12] (Figure 1). Due to the lack of Mesozoic and Paleozoic fossil records, the origin of the two plant groups, Ophioglossales (ophioglossoid ferns) and the Psilotales, is still unclear, and whether the morphological differences are caused by independent origins or transformations that postdate a shared origin is unknown [12,13]. Schneider proposed that the body plan of whisk ferns may have evolved through a secondary reduction from the common ancestor with Ophioglossales rather than primary absence [12]. Nonetheless, the key developmental mechanism of whisk ferns in leaves and roots is poorly understood and based on limited evidence.

Ophioglossales and whisk ferns are phylogenetically close and both have simplified leaf and root structures [12,14], which makes them good models for studying species with distinct organ structure. Comparative analysis of gene families, especially transcription factors (TFs) and transcriptional regulators (TRs), which play an important role in plant growth and organ development, has been widely used in identification of genes of interest [15]. In addition, as an important component of comparative genomics, phylogenetic analysis could provide insight into the functional differentiation of genes; for example, the WUS homeobox-containing (WOX) gene family and the knotted1-like homeobox (KNOX) gene family function in root initiation and leaf development, respectively [16,17,18,19]. Furthermore, selective pressure analysis of orthologous genes could provide important information on how whisk ferns proceed with structural simplifications by adaptive evolution under epiphytic environments.

In this study, we conducted phylogenetic analyses on single-copy protein-coding genes from 11 species, including *Physcomitrium patens, Selaginella moellendorffii, Arabidopsis thaliana, Equisetum arvense, Angiopteris fokiensis, Ophioglossum pendulum, O. vulgatum, Botrychium japonicum, Psilotum nudum, Tmesipteris tannensis* and *Dipteris chinensis*. By conducting comparative and evolutionary transcriptomics analyses, including the detection of lineage-specific gene families, genes under positive selection on different branches, TF/TR variation and phylogenetic reconstruction of gene families, we aimed to acquire insights into the genetic basis of adaptive evolution to epiphytic environments and of the highly reduced body plans of whisk ferns at the genomic level.

## 2. Results

### 2.1. Assessment of Transcriptomes and Homolog Clusters

In total, genomic data were collected for 11 species, of which three were downloaded from the phytozome (v13) database, five were retrieved from Sequence Read Archive (SRA), and the remaining three transcriptomes (*D. chinensis*, *O. pendulum* and *O. vulgatum*) were newly sequenced in this study (Appendix A). The GC content of each sample was 44.11–52.63% with a contig N50 between 1056–1698 bp. The completeness of acquired data was assessed by estimating the coverage of the gene space based on gene mapping to the core plant genes’ datasets, all of which supported a high quality of the assemblies: (1) 70.2–99.7% of conservatized complete genes, (2) 55.8–99.5% of complete single-copy genes. Only single-copy genes between 0.1–16.5% were classified as missing, indicating fine coverage and high quality of the peptides for these species. The number of non-redundant protein sequences in each species ranged from 18,850 in *O. pendulum* to 50,890 in *B. japonicum*. A total of 348,947 non-redundant protein sequences were sorted into 30,051 orthogroups, which covered 82.3% of genes in all non-redundant sequences. Among the 30,051 orthogroups, 65 were single-copy orthogroups.

### 2.2. Comparative Transcriptome Analyses and Functional Enrichment

Comparison of gene families between whisk ferns (*P. nudum* and *T. tannensis*) and ophioglossoid ferns (*O. pendulum*, *O. vulgatum* and *B. japonicum*) revealed 6998 core orthologs (shared by all five species) (Figure 2). In contrast, 1323 and 1588 gene families were lineage-specific to *P. nudum* and *T. tannensis*, respectively. GO functional annotation and enrichment were conducted on the lineage-specific gene families. The top 20 enriched GO terms were selected and are shown in Appendix A. These GO terms constituted genes for binding proteins (e.g., GO:0003730, GO:2001070, GO:0001871) and transmembrane transporters (e.g., GO:0080161, GO:0015171, GO:0010328), which potentially play important roles in the defense of limited water and poor nutrient supply of epiphytes.

### 2.3. Phylogeny Reconstruction and Analysis on TF/TR Variation Related to Morphological Development

For each of the 65 single-copy gene families, alignment was conducted on protein sequences from the 11 plant species. The resultant 65 alignments were concatenated to generate a supermatrix, which was used for phylogeny analysis. The maximum likelihood species tree illustrated a well-resolved phylogeny with most nodes having a 100% bootstrap value (BSV) except two nodes with a BSV at 99% and 76%, respectively (Appendix A). As expected, the whisk fern clade consisting of *P. nudum* and *T. tannensis* was clustered with the ophioglossoid fern clade consisting of *O. pendulum*, *O. vulgatum* and *B. japonicum*.

Based on the constructed species tree, a computational analysis of TF/TR family sizes was conducted (Appendix A) to study the expansion and contraction of TF/TR gene families involved in morphological development during the evolution of whisk ferns and their related species (Figure 3 and Appendix A). The result showed that 26 and 47 TF/TR gene families were under expansion in *P. nudum* and *T. tannensis*, respectively, compared to 12, 10 and 71 in *O. pendulum*, *O. vulgatum* and *B. japonicum*, respectively. On the other hand, 37 and 20 TF/TR gene families were under contraction in *P. nudum* and *T. tannensis*, respectively, compared to 44, 43 and 9 in *O. pendulum*, *O. vulgatum* and *B. japonicum*, respectively. To explore the most significantly changed TF/TR involved in morphological development, TF/TR gene families with a rapid evolution were selected and are listed in Table 1. Some TF/TR gene families involved in morphological development were found to reduce in family size in whisk ferns. For example, the GRF gene family, whose expression was strongly upregulated in actively growing and developing tissues such as shoot tips, flower buds and roots, had significant contraction in both *P. nudum* and *T. tannensis*. As another example, the MADS-MIKC gene family, type II of MADS-box genes in plants, was reduced by five in *P. nudum*. Interestingly, the family sizes of GRF and MADS-MIKC were also reduced in *O. pendulum*.

### 2.4. Phylogenetic Analysis of WOX and KNOX Genes

Whisk ferns are characterized by rootless and leaf-like appendages, and applying phylogenetic strategies, especially from sequences of gene families involved in root initiation and leaf development, will help us understand the evolutionary history of the root and leaf. Based on the sequences of WOX and KNOX proteins, evolutionary trees of multiple land plants including whisk ferns were constructed (Figure 4). The phylogeny of WOX genes was divided into three subfamilies, the ancient-clade-WOX (AC-WOX), the WUS-clade-WOX (WC-WOX) and the intermediate-clade WUSCHEL-RELATED HOMEOBOX (IC-WOX). In the WC-WOX subfamily, the homologous genes of whisk ferns (*P. nudum*) and ophioglossoid ferns (*B. japonicum* and *O. vulgatum*) were clustered into a clade (Figure 4A). According to previous studies, the plant KNOX proteins could be divided into three subfamilies, Class I, Class II and KNATM [20]. The KNOX tree generated by this study also had three branches representing the three subfamilies (Figure 4B). The homologous genes of whisk ferns (*P. nudum*) and ophioglossoid ferns (*O. vulgatum*) were clustered into a clade in the Class I subfamily (Figure 4B).

### 2.5. Genes under Positive Selection Detected in Whisk Ferns

To check whether *P. nudum* and *T. tannensis* had the same adaptive evolution, genes under positive selection were detected in the branch of *P. nudum* and the branch of the common ancestor of *P. nudum* and *T. tannensis* (Figure 5). Using the branch-site model of PAML, positive selection was detected in 803 single-copy orthologs. When *P. nudum* was labeled as a foreground alone, positive selection was detected in 74 genes. Among these 74 genes, 62 were annotated, with most of them carrying functions involved in transmembrane transporter activities and stress response (e.g., sodium/hydrogen exchanger and heat shock proteins) (Appendix A). In the common ancestral branch of *P. nudum* and *T. tannensis*, positive selection was detected in 79 genes. Among the 79 genes, 68 were annotated, with most of them carrying functions involved in transmembrane transporter activity and stress response as well (e.g., BT1 protein, Mpv17/PMP22 protein and proteins for chromosome structure maintenance) (Appendix A). Though the numbers and functions of positive selection genes were not identical, the majority of these genes were involved in cellular response to water and nutrient deficiency caused by the epiphytic environment, secondary wall biosynthesis and DNA damage repair.

## 3. Discussion

Early understanding of the origin of whisk ferns was based on the well-preserved fossils of rhyniophytes, in which whisk ferns were regarded as direct descendants of the rhyniophytes due to their root-less and leaf-less morphologies [21,22]. However, both DNA barcoding and phylogenomic analyses indicated that whisk ferns are sisters of Ophioglossales [14,23]. In this study, we performed analyses on comparative transcriptomics and positively selected genes to illustrate the adaptive evolution of whisk ferns in epiphytic environments and provided insights into the origin of the simplified root and leaf of whisk ferns.

Significant numbers of unique genes in whisk ferns (*P. nudum* and *T. tannensis*) were enriched in the GO terms’ “binding proteins”, such as hormone binding, auxin binding and kinase binding, as well as “transmembrane transporter activity”, such as auxin transmembrane transporter activity, inorganic anion transmembrane transporter activity and phosphate ion transmembrane transporter activity (Appendix A). Previous studies indicated that whisk ferns are often epiphytic, with rhizomes having symbiotic mycorrhizal associations with other plants [21,24]. Due to the arboreal habitation and direct interaction with the atmosphere interface, epiphytes are subjected to multiple abiotic and biotic stresses [25,26]. For example, living in the canopy environments, the major challenges to the survival and growth of epiphytes are water deficit and obtaining nourishment directly from the environment [26,27]. Thus, genes related to transmembrane transporter and binding protein activities might contribute to adaptive evolution for water deficit stress, nutritional stress and other abiotic as well as biotic stress.

Constructing a stable phylogenetic topology structure is the basis for exploring how plant evolution related to morphological innovations was driven by alterations in the TF/TR gene families [28]. In this study, we performed a reconstruction of TF/TR to identify TF/TR genes that had changed corresponding to morphological development in whisk ferns and ophioglossoid ferns. Our results showed that the family sizes of some TF/TR related to morphological development (e.g., the GRF family and the MADS-MIKC family) were reduced in epiphytic whisk ferns (*P. nudum* and *T. tannensis*) and epiphytic *O. pendulum* (Table 1). Therefore, we hypothesized that the root-less and leaf-less whisk ferns and the simplified ophioglossoid ferns had a similar adaptive evolutionary trajectory that could be traced. Of the TF/TR genes with rapid evolution, the GRF family, one of the plant-specific TF families, plays an important regulatory role in cotyledon and leaf growth [29,30]. In *Arabidopsis*, overexpression of two of the nine GRFs resulted in larger leaf size; in contrast, insertional null mutations in three GRFs resulted in smaller leaves and cotyledons due to a decrease in cell size [30]. MADS-MIKC, type II of the MADS-box TF family, has been divided into a range of subfamilies such as ANR1, AGL12, SVP, TM3 and FLC [31,32]. Specifically, the AGL12 genes involved in root cell differentiation and the ANR1 genes involved in regulating lateral root development might have experienced a reduction in epiphytes compared to terrestrials [33,34]. We also found significant differences between whisk ferns and ophioglossoid ferns in gene families including Dof, GATA-Tify and BES1. It is worth noting that a similar adaptive evolution mechanism for root and leaf reduction in whisk ferns and ophioglossoid ferns does not imply that all TF/TR genes had the same evolutionary trajectories.

Gene expression and protein functional studies indicated that the WOX family members fulfilled specialized functions in the processes of plant root formation [35]. IC-WOX and WC-WOX were considered to be important participants in the process of root organogenesis in the common ancestor of ferns and seed plants [5]. In our analysis, homologous WOX genes were clustered with whisk ferns (*P. nudum*) and ophioglossoid ferns (*B. japonicum* and *O. vulgatum*) in a clade in the WC-WOX subfamily, suggesting the conservation of genes related to root formation in whisk ferns and ophioglossoid ferns, and supporting the hypothesis that roots of whisk ferns and ophioglossoid ferns had a common origin. The KNOX family, including three subfamilies, Class I, Class II and KNATM, is a group of TFs that play important roles in leaf and meristem development [17,20]. In this study, phylogeny based on the KNOX gene family gathered the simple leaf-shaped whisk ferns and ophioglossoid ferns together and clearly distinguished them from other plants with compound leaves such as *A. fokiensis* and *A. thaliana* (Figure 5B), which also indicates that the evolution of the KNOX family was conserved in whisk ferns and ophioglossoid ferns. In general, the evolutionary conservation of WOX and KNOX elucidates a common origin of whisk ferns and ophioglossoid ferns.

We hypothesized that adaptive evolution to epiphytism increased tolerance to water and nutrient deficits. Thus, genes under positive selection were detected in the ancestral branch of whisk ferns (*P. nudum* and *T. tannensis*) and the branch of *P. nudum*, to reveal whether adaptive evolution to epiphytism occurred within the two branches. Positive selection was detected in 79 and 74 genes in the branches of whisk ferns and P. nudum, respectively. Some of the annotated genes identified from the two branches are involved in transmembrane transporter activity and stress response (Appendix A). Whisk ferns inhabiting epiphytic and exposure environments could be exposed to heat stress, water and nutrient deficits. In this way, genes subjected to positive selection and carrying transmembrane transporter activities might contribute to adaptive evolution to the epiphytic environment. In addition, several genes under positive selection were commonly detected from both the ancestral branch of whisk ferns and the branch of *P. nudum*. These genes are also functional in response to epiphytic environments. For example, the UDP-glycosyltransferase has been found to play an important role in cell wall pentose biosynthesis and secondary wall biosynthesis [36,37], which helps rhizome and aerial branch development to adapt to specific epiphytic habitats.

Naturally, it cannot be denied that there are still some limitations in this research, and further research and discussion are needed. For example, this paper only considers the expressed genes for the species whose transcriptomes were sequenced, but in real life, RNAseq data could very easily miss genes if the RNA for a given gene is not expressed under the growth conditions or in the tissues used for RNA analysis. In addition, although one replicate per sample is reasonable for phylogenetic analysis in this research, it could result in incomplete or missing data for comparative transcriptome analysis. Moreover, in the absence of other valid validations, there is still uncertainty about GO annotations of gene functions with reference to EggNOG database. Nevertheless, our findings still preliminarily illustrate adaptive evolutionary patterns of whisk ferns in leaf-less and root-less species. Therefore, more research is still needed, and more genomic data needs to be collected to establish a trend given the uncertainties in the experiments.

## 4. Materials and Methods

### 4.1. Transcriptome Data Collection and BUSCO Analysis

Transcriptome or genome data from 11 species, including whisk ferns and ophioglossoid ferns, along with other related species that occupy distinctive phylogenetic positions, were obtained as described below (Appendix A). The whole protein sequences of *P. patens*, *S. moellendorffii* and *A. thaliana* were downloaded from the Phytozome (v.13) database [38], and the raw reads of the transcriptomic sequences of *E. arvense*, *A. fokiensis*, *P. nudum*, *T. tannensis* and *B. japonicum* were downloaded from the National Center for Biotechnology Information (NCBI) database [23,39]. The other three field-collected adult species, including terrestrial *D. chinensis*, epiphytic *O. pendulum* and terrestrial *O. vulgatum,* were newly sequenced in this study. For transcriptome sequencing, the total plant transcriptomic RNA was isolated from mixed fresh tissues of leaf, stem and rhizome using the Tiangen Polysaccharide&Polyphenolics-rich RNAprep Pure Plant Kit. Illumina library preparation, and RNA-sequencing was performed on the Majorbio next-generation sequencing (NGS) platform (Shanghai, China). To standardize, all the raw reads were cleaned using Trimmomatic (v.0.36) [40] and de novo assembled using Trinity (v.2.6.6) [41]. Coding sequences (CDSs) of the transcripts and amino acid sequences of the encoded proteins were predicted by TransDecoder (v.5.50) (https://github.com/TransDecoder, accessed on 10 August 2021) with default parameters. The cd-hit (v.4.6.2) [42] software was used to remove redundantly assembled transcripts with a sequence identity threshold at 98% and a minimum coverage at 90%.

The genomic assembly completeness of the 11 species in total were assessed by estimating the coverage of the gene space based on blast against the core plant homologous gene database (www.orthodb.org, accessed on 5 August 2021).

### 4.2. Phylogenetic Analysis

The OrthoFinder (v.2.2.6) [43] software was used to cluster transcripts into orthologous genes (OGs), which infer core-orthogroups based on all-against-all BLASTP (v.2.9.0+) [44] searches with an e-value cutoff at 10^−5^. In total, 65 single-copy OGs were identified. Orthologs in each of the 65 single-copy OGs were aligned using MAFFT (v.7.471) [45]. For the concatenation analysis, the Utensils phylogenetic tool (https://github.com/ballesterus/Utensils, accessed on 21 October 2021) was used to generate a supermatrix from the alignments. A phylogenetic tree was reconstructed from the supermatrix using IQ-TREE (v.2.0.3) [46] with 1000 bootstrapping repeats and using -m MFP parameter for automatically extended model selection followed by tree inference. The resultant maximum likelihood tree was subjected to subsequent analyses.

### 4.3. Comparative Transcriptome Analysis

Using the OrthoVenn2 web platform [47], all protein sequences from the 5 species were compared. The output of OrthoVenn2 was an interactive occurrence pattern table, and within the occurrence results, the functional annotations and summaries of the disjunctions and intersections of clusters between the species were displayed through a Venn diagram. The gene ontology (GO) term enrichment was performed using the GO enrichment tool included in the OrthoVenn2 platform. Additionally, the iTAK (v.18.12) program [15] was used for prediction and classification of plant TFs and TRs from all the protein sequences. Based on the constructed evolutionary tree, expansion and contraction of TFs and TRs relative to the most recent common ancestor were analyzed using the CAFÉ (v.4.2.1) program [48]. The value of the birth and death parameter (λ) was 0.00253807 and the *p*-value was 0.01.

### 4.4. Identification and Gene Tree Reconstruction of the WOX and KNOX Families

Two methods, BLASTP based on the protein homology alignment and hidden Markov model (HMM) search based on the HMM files, were used to identify the WOX and KNOX proteins. BLASTP and HMM search was conducted using local BLAST (v.2.9.0+) and HMMER (v.3.2.1), respectively. *Arabidopsis* WOX and KNOX proteins were obtained from the TAIR database (https://www.arabidopsis.org, accessed on 13 November 2021) and used to construct a local BLAST database, against which BLASTP was conducted using the protein sequences from the 11 species as queries with a cut-off e-value equaling 10^−5^. Domain PF00046 (homeobox domain) from the Pfam database (http://pfam.xfam.org, accessed on 13 November 2021) was used for HMM search with an e-value at 10^−5^. Then, the NCBI Conserved Domain Database and SMART (the Simple Modular Architecture Research Tool; http://smart.embl-heidelberg.de, accessed on 13 November 2021) were used to confirm the putative proteins. To study the evolutionary relationship between species based on the sequences of WOX and KNOX proteins, an evolutionary tree was constructed using sequences of A. thaliana and related species by the PhyloSuite (v.1.2.2) [49] platform, with the following built-in software with default parameters: MAFFT for the alignment of all identified sequences; ModelFinder for the automatic detection of evolutionary models; and IQ-TREE for the construction of unrooted maximum-likelihood phylogenetic trees.

### 4.5. Detection of Genes under Positive Selection

Presence of positive selection was detected using the CodeML program of the PAML software package [50], which implements the branch site model. CodeML computes ω, the ratio of nonsynonymous over synonymous substitution rates (dN/dS). In this study, two models were fitted and tested based on the null and alternative hypotheses of sequence evolution. The null hypothesis assumes that the gene evolved neutrally and ω is expected to have a value of 1 in all branches (null model: model = 2, NSsites = 2, fix_omega = 0, omega = 1), while the alternative hypothesis assumes that the gene is under positive selection and ω value is greater than 1 in the foreground branch (model A: model = 2, NSsites = 2, fix_omega = 1, omega = 1). For each of the two models, goodness of fit between the expected and observed ω was determined by the *p*-values of chi-squared test. Based on their annotation, the positively selected genes were assigned to a Cluster of Orthologous Groups (COG) category by searching the COG EggNOG database [51], and the enrichment analysis was performed using the OmicShare tool, a free online platform for data analysis (https://www.omicshare.com/tools, accessed on 8 December 2021).

## 5. Conclusions

In this study, we performed analyses on comparative transcriptomics and positive selection of genes to explore the adaptive evolution of whisk ferns. We identified gene families specific to whisk ferns and genes showing signs of adaption to epiphytic environments. The gene family size of TF/TR related to morphological development (e.g., the GRF family and the MADS-MIKC family) was reduced in epiphytic whisk ferns (*P. nudum* and *T. tannensis*) and epiphytic *O. pendulum*. Phylogenetic analysis revealed that both the WOX and KNOX gene families had a conserved evolutionary history in whisk ferns and ophioglossoid ferns. Genes under positive selection were detected in the ancestral branch of whisk ferns and the branch of *P. nudum*, and these genes were mainly involved in the stress response (e.g., heat shock proteins and proteins for chromosome structure maintenance) and transmembrane transporter activities (e.g., BT1 protein and sodium/hydrogen exchanger), which suggested that the positively selected changes on these genes might be associated with adaptation to epiphytic habitats and nutrient deficiency. Overall, we proposed that the simplified and reduced leaves and roots in whisk ferns are the result of reduction from the common ancestor of whisk ferns and ophioglossoid ferns, rather than an independent origin, and that the adaptation to the epiphytic environment in these ferns could be assessed at the gene level.

## Figures and Tables

**Figure 1 plants-11-01198-f001:**
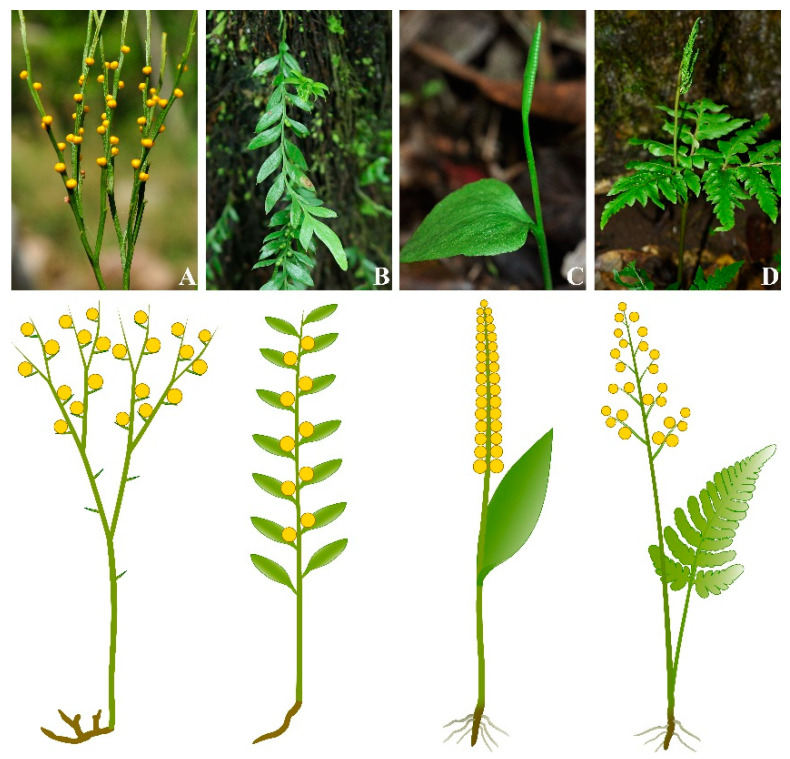
The distinct morphologies of whisk ferns and ophioglossoid ferns. (**A**,**B**), the whisk ferns, *Psilotum nudum* (**A**) and *Tmesipteris tannensis* (**B**) are characterized by the lack of roots, the reduction of leaves, the leaf-like appendages and the absorptive rhizoids. (**C**,**D**), the ophioglossoid ferns, *Ophioglossum vulgatum* (**C**) and *Botrychium japonicum* (**D**), are characterized by simple or pinnatifid leaves and a reduced root system.

**Figure 2 plants-11-01198-f002:**
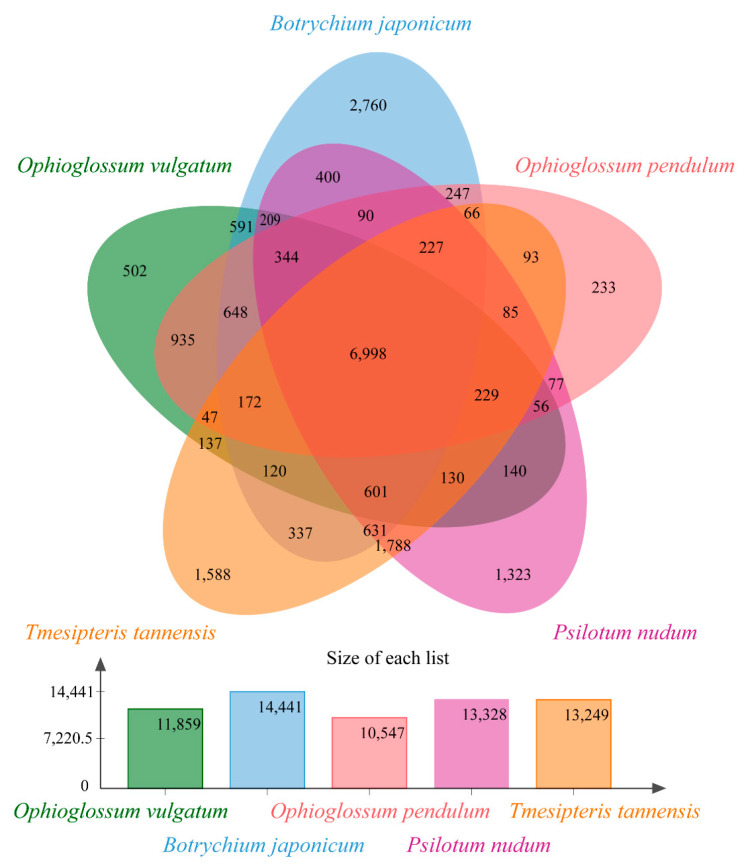
Venn diagram showing unique and shared gene families among members of whisk ferns (*Psilotum nudum* and *Tmesipteris tannensis*) and ophioglossoid ferns (*Ophioglossum vulgatum*, *O. pendulum* and *Botrychium japonicum*). The number in each region represents the number of gene families belonging to that category.

**Figure 3 plants-11-01198-f003:**
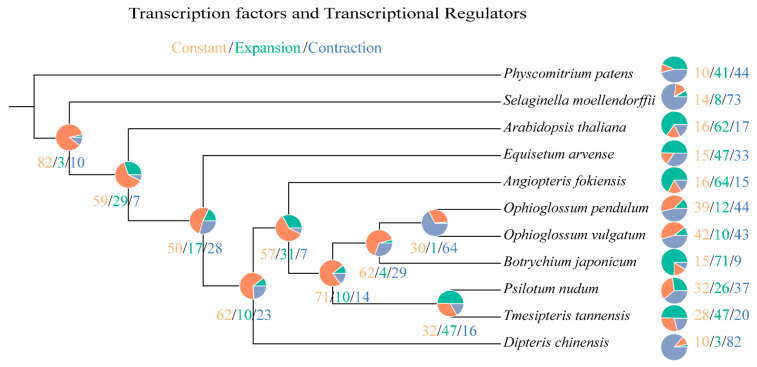
Phylogenetic tree showing the evolution of TF/TR gene family sizes in 11 plant species. The whisk fern clade consisting of *Psilotum nudum* and *Tmesipteris tannensis* is clustered with the ophioglossoid fern clade consisting of *Ophioglossum pendulum*, *O. vulgatum* and *Botrychium japonicum*.

**Figure 4 plants-11-01198-f004:**
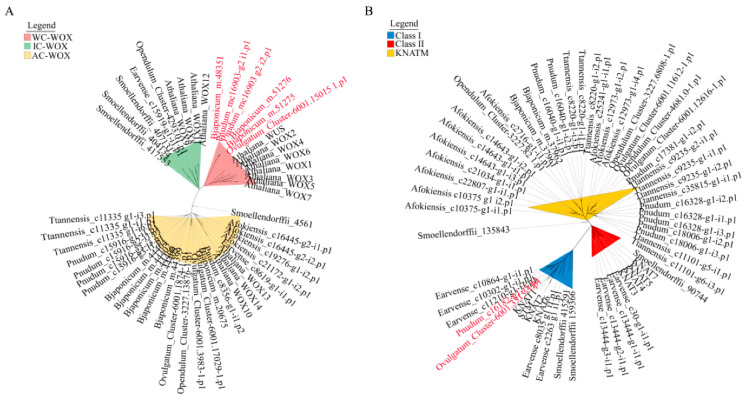
Grouping of WOX and KNOX homologs using the neighbor-joining method. (**A**) A collection of 50 WOX proteins from 9 distinctive plant species. The WC-WOX sequences of whisk ferns and ophioglossoid ferns are highlighted. (**B**) A collection of 58 KNOX proteins from 9 distinctive plant species. The class I sequences of whisk ferns and ophioglossoid ferns are highlighted.

**Figure 5 plants-11-01198-f005:**
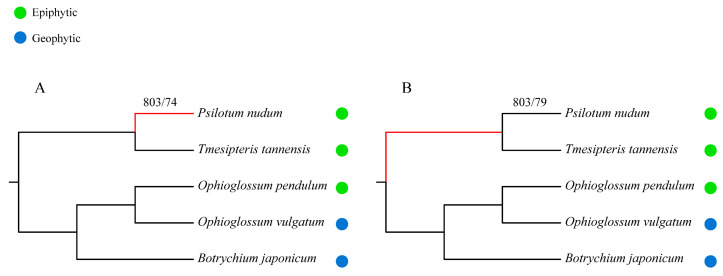
The phylogenetic relationships of whisk ferns and ophioglossoid ferns used to detect genes under positive selection. (**A**) Selection pressure analysis where *Psilotum nudum* was labeled as the foreground. (**B**) Selection pressure analysis where the common ancestral branch of whisk ferns was labeled as the foreground. Branches in red color indicate foreground branches for detection of genes under positive selection. The color of the circle represents the ecological habit of each species, with green for epiphytic habit and blue for geophytic habit.

**Table 1 plants-11-01198-t001:** A summary of rapidly evolved TF/TR families identified by CAFÉ.

Species	Rapidly Evolved TF/TR Families
*Physcomitrium patens*	null
*Selaginella moellendorffii*	null
*Arabidopsis thaliana*	MADS-MIKC [+25 *]
*Equisetum arvense*	AUX-IAA [+14 *], HB-BELL [+12 *], SBP [+16 *], GATA-Tify [+17 *]
*Angiopteris fokiensis*	BES1 [+8 *], HD-ZIP [+16 *], Jumonji [+12 *], MADS-MIKC [+28 *], TRAF [+25 *]
*Tmesipteris tannensis*	Dof [+7 *], GATA-Tify [+9 *], BES1 [−4 *], GRF [−3 *]
*Psilotum nudum*	GRF [−5 *], Dof [−8 *], MADS-MIKC [−5 *]
*Botrychium japonicum*	BES1 [+6 *], GARP-G2-like [+15 *], HB-other [+11 *], IWS1 [+10 *], MYB-related [+17 *], TRAF [+17 *]
*Ophioglossum vulgatum*	null
*Ophioglossum pendulum*	GRF [−3 *], HB-other [−6 *], MADS-MIKC [−5 *]
*Dipteris chinensis*	Dof [−15 *], GARP-G2-like [−17 *], HB-HD-ZIP [−14 *], Jumonji [−11 *], MYB-related [−22 *], SNF2 [−20 *], TRAF [−14 *]

Note: The [+number] indicates the number of gained genes, the [−number] indicates the number of lost genes, and * indicates a rapid change compared to other changes in that gene family in other lineages. The change information of whisk ferns and ophioglossoid ferns is highlighted.

## Data Availability

The raw sequence data reported in this paper have been deposited in the SRA under Bioproject number PRJNA762181.

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
