# Peer review of "Analysis of Comparative Transcriptome and Positively Selected Genes Reveal Adaptive Evolution in Leaf-Less and Root-Less Whisk Ferns"

_plants, 2022, doi:10.3390/plants11091198_

Round 1
Reviewer 1 Report
Journal: Plants
Title: Analysis of comparative transcriptome and positively selected genes reveal adaptive evolution in leaf-less and root-less whisk ferns.
Authors: Zengqiang Xia, Li Liu, Zuoying Wei, Faguo Wang, Yuehong Yan
The whisk ferns (Psilotum spp.) are a peculiar group of plants comprises two species: more common Psilotum nudum and the lesser known P. complanatum. Recent study by Shen et al. (2018) revealed that Psilotales (whisk ferns), together with Ophioglossales (moonworts) and Marattiales (king ferns) form a monophyletic clade which is sister to true (leptosporangiate) ferns. The research was yielded from a large-scale phylogenomic analysis based on a high-quality RNA-seq dataset covering 69 fern species.
Interestingly, what is characteristic for whisk ferns, they lack both roots and leaves, which were probably lost by evolutionary reduction. In the revised manuscript, the authors conclude that reduced leaf and root system is the result of reduction from the common ancestor of whisk ferns and ophioglossoid ferns, rather than an independent origin. The conclusions result from phylogenomic and comparative transcriptomic analyses base on 11 plant species including Psilotum nudum. It is important, that three of the species were newly sequenced in this study (D. chinensis, O. pendulum and O. vulgatum), and eight were de novo assembled transcriptomes.
I found only a few reports on the molecular analysis of this species in the available literature. Therefore, presented experiments seems to be original and innovative. In my opinion quality of written English is suitable for publication, but I am not qualified to decide on this issue. Methods used in manuscript were described precisely. To sum up, this manuscript could be acceptable for publication in Plants after minor corrections.
The manuscript is written with some of small mistakes:
- Authors should check carefully spaces in all text.
- Table 1- all the text should be in black.
- References – position 17 and 39, lack of dot after ‘Biol’ in the name of the journal.
- I suggest to use larger font for to describe the diagram (Fig. 1) and phylogenetic trees (Fig. 4). The names and numbers (a bar graph) are illegible.
- Author Contributions: Please, specify more exactly the role of Faguo Wang in the research.
Author Response
Response to Reviewer 1 Comments
Dear reviewers:
On behalf of my co-authors, we thank you very much for giving us an opportunity to revise our manuscript, we appreciate editor and reviewers very much for your positive and constructive comments on our manuscript entitled “Analysis of comparative transcriptome and positively selected genes reveal adaptive evolution in leaf-less and root-less whisk ferns” (Manuscript ID: plants-1651511). We have studied your comments carefully and have tried our best to revise our manuscript according to the comments. The main corrections in the paper and responds are as flowing:
Point 1: Authors should check carefully spaces in all text.
Response 1: Thank you for your detatiled review, we have checked and revised spaces in the full manuscript in detail.
Point 2: Table 1- all the text should be in black.
Response 2: All text in table 1 has been modified to black.
Point 3: References – position 17 and 39, lack of dot after ‘Biol’ in the name of the journal.
Response 3: I have added dot after ‘Biol’ and checked the format of other references in detail.
Point 4: I suggest to use larger font for to describe the diagram (Fig. 1) and phylogenetic trees (Fig. 4). The names and numbers (a bar graph) are illegible.
Response 4: Thanks for your suggestion, we have used larger font in Fig.1 and Fig.4 so that they can be viewed clearly.
Point 5: Author Contributions: Please, specify more exactly the role of Faguo Wang in the research.
Response 5: F.W. collected the plant material and F.W. modified the manuscript have been added to section “Author Contributions”. Thank you for your reminder.

Reviewer 2 Report
Plants Manuscript #: plants- 1651511
Authors: Z. Xia et al.
Title: Analysis of comparative transcriptome and positively selected genes reveal adaptive evolution in leaf-less and root-less whisk ferns.
The authors have presented a combination of genomic (transcriptomic) and proteomic (predicted protein sequence) data from different ferns to assess the evolution of predicted transcription factor (TF) and Transcription Regulatory (TR) genes that might relate to the anatomy (root-less and leaf-less) and epiphytic habitat for the primitive whisk ferns. They mainly focus on two whisk ferns (Psilotum nudum and Tmesipteris tannensis) and compare these to the Ophioglossoid ferns (Ophioglassum vulgatum and Ophioglassum pendulum), a phylogenetically very close sister-group to Whisk ferns. They also include more distantly related ferns and angiosperm plants as “out group” controls for their comparisons. The species they used to test and compare to Whisk Ferns is appropriate.
There are, however, a combination of issues with the manuscript regarding the provided data, explanation of data and methods (mainly lack of key standard information about RNAseq experiments and data), lack of actual RNA expression data, wording in the writing for some sections, some missing citations/references, and stated conclusions that - at times – appear to go beyond what the data can support and therefore are overly speculative. Extensive additional information about the RNAseq methods, plant growth information and tissues/organs used for RNAseq analysis, RNAseq QC data, possibly additional data, and changes to writing/wording to reflect the more speculative conclusions are needed before I could support the publication of the manuscript. Below are my general and specific concerns, with some suggested edits/changes for certain issues.
General scientific concerns that need to be addressed:
First, and most significantly, there needs to be much additional information provided about the RNAseq method and experimental design to know how to interpret the provided data. The authors provided insufficient amounts of information about the RNAseq method. In general, RNAseq data should follow the accepted guidelines to what information needs to be provided in a paper as is described in the commonly cited/used “ENCODE Guidelines and Best Practices for RNA-Seq, 2016” (https://www.encodeproject.org). At a minimum this should include: (1) how many replicate RNA samples from each species and tissue/organ type were used (three independent RNA isolated replicates and libraries prepared for RNAseq is the general minimum provided/expected), (2) exact growth conditions for the ferns (soil or epiphytic growth; temperature, light conditions, etc…), (3) how old were the plants for RNA isolation, and (4) what tissues/organs were collected for RNA isolation. Beyond that, information about RNAseq data quality (QC) related to sequencing depth (minimum of 10 million, 30 million, 100 million aligned paired reads?) and read statistics and quality scores also need to be provided. I did look up the NCBI-SRA sources of uploaded data (as cited in Supplemental S1 Table), which were there, but the needed information about the RNAseq method (asked for above) were not there, that I could find. Without this information, it is very difficult to impossible to know how much to trust the RNAseq data and this puts into question any conclusions from these data.
Second, the authors provided no information about differential gene expression (DGE) or RNA isoform/splicing variants data analysis from the RNAseq data across the species. This seems odd, for those are often the common uses for RNAseq data. I fully realize the authors were using the RNAseq data to generate predicted coding sequences (cds) to then predict hypothetical encoded proteins for the comparisons between different fern species. But, there is the potential for much additional information/data around DGE and RNA isoforms/splicing that is not mentioned or characterized here. This does relate to the question of speciation and evolution of plants/ferns. For example, speciation can result, in part, from sequence changes in promoters/regulatory sequences and not always just from changes in coding sequences. Changes in promoter or regulatory sequences can alter RNA expression over development, environmental conditions and this could impact plant morphology and evolution. Thus, genetic changes that drive speciation are not always just from gain/loss of genes or changes that alter sequence of encoded proteins. A well-known example of this is the domestication of Zea mays from ancestral teosinte in which changes in the promoter region of the teosinte branched1 (tb1) gene affects difference in plant morphology (F. Tian et al., 2009 PNAS 106, 9979-9986.). The RNAseq data might miss such key genetic changes.
Third, and this relates to the item one above. The authors use the transcriptomic data (specifically, the estimated coding sequences) to predict hypothetically encoded proteins. These hypothetical protein sequences were then used for the phylogenetic and gene select analysis. Since they are using protein sequences, formally this is proteomic data (not transcriptomic data) and the text (headings and body of text) should reflect this. Also, since the hypothetically predicted protein sequences from RNAseq data are being used for the “gene” comparisons, these data and this approach could very easily miss genes/protein if the RNA for a given gene is not expressed under the growth conditions or in the tissues/organs used for RNA analysis. If that is the case, it would not show up on their data. This goes back to my first general concern from above, and it being critical to know how the plants were grown, what age, and what tissues/organs were used for RNA isolation to generate the RNAseq data. A hypothetical example of how important this might be relates to Table 1 in the manuscript in which the authors state that in P. nudum the predicted GRF gene family has contracted (lost) 5 predicted GRF genes, and the authors use these as date to conclude there was a rapid evolutionary change in the important GRF regulatory genes. However, what if it is simple due to the fact that those genes in P. nudum were, by chance, not expressed as RNA under the growth conditions or tissues/organs used for RNAseq experiments. It would appear as if that gene (or those genes) were lost, but it might simple be the genes are present but not expressed in the conditions or tissues used here. This could lead to an erroneous conclusion. Or, alternatively, what if the RNA is expressed but the RNA splicing is different in this species or under these conditions such that the hypothetical protein is sufficiently different that it no longer is predicted by Gene Ontology (GO) identifying software to match a GRF gene family. But, it might still be there and function, just in an unusual way. Again, this is hypothetical, but based on the type of data and lack of information about RNAseq method and data quality, as I see it, this possibility cannot be ruled out.
Fourth, there appears to be inconsistency between data in the summary Table 1 and what is shown in Supplementary S2 Table. Specifically, Table 1 states for P. nudum there is a loss of 5 (“-5”) GRF genes as well as a loss of 5 (“-5”) MADS-MIKC genes, and this is used as evidence to show a rapid evolution for these important plant TF/TR genes that might correlate with the root-less and leaf-less anatomy of the Whisk Ferns. However, Supplemental S2 Table lists how many of the GRF genes are in P. nudum, and it indicates there are “10” GRF genes in P. nudum (Whisk Fern) as compared to ten other fern / plant species that have an estimated 0 to 9. Thus, P. nudum has the most GRF genes (10) of the eleven species tested. This not only does not match a loss of 5 (“-5”) but in fact it has the most. Please explain this inconsistency. Note, I did not go through and check all of the 95 different annotated TF/TR gene families across all eleven species to see if these too were inconsistent with the Table 1 data. These too need to be checked and adjusted or explained.
Final general comment. The conclusions stated are, in my opinion, at times very speculative with regard to whether the changes in predicted genes/encoded proteins for TF/TR are present or absent (and based unusually and possibly incorrectly on RNAseq, as discussed above) and then to use that as a bases to suggest those are the causes of the root-less and leaf-less anatomy. In fact, data in Figure 5 seems to argue against this, to some extent. The figure presents data on genes that show positive selection in the epiphytic Whisk Fern P. nudum. However, if the positively selected 74 of 803 genes (as stated for P. nudum) are leading causes for the epiphytic growth in P. nudum, then it would also predict they would be selected for in other epiphytic ferns, including O. pendulum. Based on the provided, that would not seem to be the case. Now, it might in fact be the case that the identified TF/TR genes and there evolution (loss and gain) are indeed directly involved the epiphytic growth and root-less/leaf-less anatomy of Whisk Ferns, but since TF/TR can vary in very subtle ways or can be used in a combinatorial way with other TF to impact transcriptional regulation, using the GO predictions alone is not sufficiently robust to have confidence these changes are the causative reason for the change in anatomy in the Whisk Fern. As I see it, the wording in the conclusions needs to reflect the uncertainty and speculator nature of these conclusions from these data.
Suggested specific edits and/or questions, ordered chronologically by Sections, Paragraphs, and Lines (“L”) in manuscript:
Lines 57-61: Related to classification and evolution of ferns, the Smith et al. “A Classification Extant Ferns” (2006, Taxon. Vol 55, pp 705-731) is often cited.
Line 78 (and other locations in manuscript including figures): Some sources now use the name Physcomitrium patens instead of Physcomitrella patens name used in the manuscript. This includes the Phytozome database of the genetic sequence data that the authors cite.
Lines 93-95: Data include both Transcriptomes and Proteomes (since used predicted protein sequences). Further, there needs some general information about RNAseq method with regard to age of plants, how grown, and what tissues/organs used. Full details need to be in the Materials and Methods (Lines 274-282), but some general information would be good to include in the Results.
Line 97: Need to make it clear the protein sequences are “predicted hypothetical” sequences from the transcriptomes and not from direct protein sequencing.
Line 99: the wording, “…from the 11 genomic data were …” is awkward and should be changed.
Figure 2: The font color used (especially yellow font) is nearly impossible to read. Change font size and/or color to be readable.
Line 129: Cites Supplemental Table S2. In that table, the authors have column heads related to predicted TF/TR genes that are not clearly defined. They need to articulate the difference between what “Expanded TF/TR” means compared to “Genes gained”. Explain the difference in these data/gene numbers. Similarly for the “Contracted TF/TF” and “Genes Lost” headings.
Lines 132 and 134: Mentions TF/TR “genes”, but I believe these really refer “gene families”. Please check and clarify, if necessary.
Lines 133-136: This relates to my general concerns above and related to my “Third” general concern and that RNAseq data (from RNA) might “miss” genes if that gene or group of genes were not expressed or at sufficiently low level under particular growth conditions, developmental stage, or tissues used that they don’t appear in the RNA data. This might give a false predict of gene lost or “contraction”.
Line 137: Cites Table 1 and claims of rapid evolution of TF/TR gene families. I already pointed out about what seems to be inconsistency in summary data in Table 1 versus full detailed data provided in Supplemental S2 Table. These need to be checked and resolved and/or explained better so that all data are consistent, and clear to readers.
L139-140: claims that for GRF gene family there is, “significant contraction in both P. nudum and T. tannensis.” Table 1 states this is a loss of 5 genes (-5) for P. nudum and loss of 3 genes (-3) for T. tannensis. It is not clear what species these are compared to to know if there is a loss or not. And, even with that claim, as mentioned above, data in Supplemental S2 Table does not seem to support that claim. Please clarify.
Figure 3: Not completely clear where the data for the provided pie charts that show gene families that appear to stay Constant, Expand, or Contract comes from. For example, the pie chart shown for the first node of the Psilotum nudum and Tmesipteris tannensis seems to match Tmesipteris tannensis exactly, when if it for the node it would seem it should be a mix or average that factors in both Psilotum nudum and Tmesipteris tannensis.
Figure 3 Legend: I would indicate that text for species and data that is highlighted is “highlighted in red font.”
Table 1: I have already pointed out what to me seems to be inconsistency in the summary data provided in Table 1 compared to the full data set provided in Supplementary S2 Table.
Line 155: Awkward wording for, “…ferns carry usefully genetic information, …” Change to improve wording.
Lines 159-162: This mentions different predicted subfamilies of WOX’s, including the AC-WOX, WC-WOX, and IC-WOX. These data come, in part, from the Supplementary S2 Table, but in that supplementary table it is only described as the HB-WOX gene family. If the AC-WOX, WC-WOX, and IC-WOX groups have been used before, please cite a reference for this. If not, then please explain the bases for how these names and implications of evolutionary timing was generated.
Figure 4: The text of fern / plant species show in both A and B are nearly impossible to read, due to both small font size and low quality/resolution. Figure would need to be redone to make text readable.
Lines 182 and 186: Cited Supplementary Table S4 and Table S5. Note, these tables both have a type in the heading on upper right side. It should be “Annotation in EggNOG”.
Lines 186-189: Claims functions related to positively selected genes in the Whisk Ferns, which as mentioned above, is based on RNAseq and protein predictions along with very general grouping into different Gene Ontology (GO) groups, shows that “the majority of these genes were involved in cellular responses to water and nutrient deficiency caused by the epiphytic environment, secondary wall biosynthesis, ….” This is way to speculative and here are no data provided or even cited related to the physiological claims made, let along that a list of genes based on hypothetical GO is what is causing this. This sentence needs to be either removed more written so it is clear that this is speculation based on very general data.
Discussion: In general, some sections are very speculative and as worded implies a stronger conclusion than the data support, similar to point made above for Lines 186-189.
Line 206-210: GO categorizing of new genes alone does not prove biochemical / physiological function. And, for some categories, such as “Kinase binding” there are so many genes/proteins that fit in this with vastly different biological functions, that it is very speculative.
Lines 232-236: Discussion about MADS-box subfamilies such as ANR1, AGL12, etc… in rice and Arabidopsis is fine, but was not made clear that these forms exist in Whisk Fern (nor were these mentioned in the Result). Further, these MADS-box proteins might have completely different functions in Whisk Fern compared to angiosperms. There was not a clear link made to Whisk Ferns, aside from MADS-box genes being present in these ferns, which is true for ALL eukaryotes (ferns to fish, plants to humans).
Lines 257-258: Figure 5 does not show a definitive correlation with positively selected genes (identified in P. nudum) with epiphytic growth since O. pendulum is also an epiphyte and seems to be outside of those positively selected gene groups.
Materials and Methods: As mentioned above in general concerns, significant more information is needed to describe how plants were grown, age of plants, what tissues/organs used, and how replicates used for RNA isolation to be used with the RNAseq method and data. Further, information about RNAseq data quality (QC) related to sequencing depth (minimum of 10 million, 30 million, 100 million aligned paired reads?) and read statistics and quality scores also need to be provided. Information consistent with the “ENCODE Guidelines and Best Practices for RNA-Seq, 2016” (https://www.encodeproject.org) should be included.
Line 302: Heading reads, “Comparative transcriptome analysis”, yet what was really used was predicted hypothetical protein sequences, making it “proteomic analysis”. Make sure terminology used matches methods.
Lines 346-350: Again, this is very speculative to claim these identified OG genes show signs of adaption to epiphytic environment. Wording and conclusions need to be consistent with what the data can support. Speculation is okay to a point in the Conclusion, but it needs to be clear and consistent with data.
Author Response
Response to Reviewer 2 Comments
Dear reviewers:
On behalf of my co-authors, we thank you very much for giving us an opportunity to revise our manuscript, we appreciate editor and reviewers very much for your positive and constructive comments on our manuscript entitled “Analysis of comparative transcriptome and positively selected genes reveal adaptive evolution in leaf-less and root-less whisk ferns” (Manuscript ID: plants-1651511). We have studied your general and specific comments carefully and have tried our best to revise our manuscript according to the comments. The main corrections in the paper and responds are as flowing:
General scientific concerns that need to be addressed:
Point 1: First, and most significantly, there needs to be much additional information provided about the RNAseq method and experimental design to know how to interpret the provided data. The authors provided insufficient amounts of information about the RNAseq method
Response 1: Thank you for this very constructive advice. We have added additional RNAseq information in the Methods section, including sequencing location and kit for RNA extraction. The quality of RNAseq data was assessed by BUSCO assessment, N50 size and GC percentage to provide reliable information on which to trust these data (Supplementary Table S1).
Point 2: Second, the authors provided no information about differential gene expression (DGE) or RNA isoform/splicing variants data analysis from the RNAseq data across the species. This seems odd, for those are often the common uses for RNAseq data.
Response 2: Thanks for your valuable advice. Indeed, RNAseq data, taking into account the consistency of RNA isolated replicates, sample age, growth conditions and tissue/organ type, is often used for differential gene expression (DGE) or RNA isoform/splicing variants analysis. However, all the published data we collected varies greatly in data types, sampling locations, and data sources. Thus, we used these data to provide insights into the adaptive evolution in leaf-less and root-less whisk ferns from a macro-perspective on the basis of phylotranscriptomics and comparative transcriptomics. At this stage, we expected to discover evolutionary traces of whisk ferns and will explore accurate DGE analysis after completing the related data in the next step.
Point 3: Third, and this relates to the item one above. The authors use the transcriptomic data (specifically, the estimated coding sequences) to predict hypothetically encoded proteins. These hypothetical protein sequences were then used for the phylogenetic and gene select analysis. Since they are using protein sequences, formally this is proteomic data (not transcriptomic data) and the text (headings and body of text) should reflect this. Also, since the hypothetically predicted protein sequences from RNAseq data are being used for the “gene” comparisons, these data and this approach could very easily miss genes/protein if the RNA for a given gene is not expressed under the growth conditions or in the tissues/organs used for RNA analysis. If that is the case, it would not show up on their data.
Response 3: This is a relatively professional advice. Though proteomic data is an alternative, in this study, the data mainly used to explore the adaptive evolution in leaf-less and root-less whisk ferns come from transcriptome sequencing. Referring to previous study such as Zhang et al., (2019) in MDPI-IJMS, we think transcriptomic data is more appropriate. For RNAseq, these data and this approach do have missing genes/proteins if the RNA for a given gene is not expressed under growth conditions or in the tissue/organ used for RNA analysis. Nevertheless, whisk ferns, a non-model plant with a very large genome size (71G ~ 150G), faces several challenges for complete genome sequencing, including the complexity of complete genomes, the sequencing costs, and the computational resources required. In addition, it has also been shown that RNAseq can still provide useful insights in adaptive evolution analysis (Qi et al., 2018).
Point 4: Fourth, there appears to be inconsistency between data in the summary Table 1 and what is shown in Supplementary S2 Table. Specifically, Table 1 states for P. nudum there is a loss of 5 (“-5”) GRF genes as well as a loss of 5 (“-5”) MADS-MIKC genes, and this is used as evidence to show a rapid evolution for these important plant TF/TR genes that might correlate with the root-less and leaf-less anatomy of the Whisk Ferns. However, Supplemental S2 Table lists how many of the GRF genes are in P. nudum, and it indicates there are “10” GRF genes in P. nudum (Whisk Fern) as compared to ten other fern / plant species that have an estimated 0 to 9. Thus, P. nudum has the most GRF genes (10) of the eleven species tested. This not only does not match a loss of 5 (“-5”) but in fact it has the most. Please explain this inconsistency.
Response 4: The significant changes (expansions or contractions) in the data in the summary Table 1 are relative to the most recent common ancestor, supported by the results of café analysis. While supplemental S2 lists the number of TF/TR based on the acquired transcriptomic/genomic data of each species, and the results are supported by the iTAK analysis.
Point 5: Final general comment. The conclusions stated are, in my opinion, at times very speculative with regard to whether the changes in predicted genes/encoded proteins for TF/TR are present or absent (and based unusually and possibly incorrectly on RNAseq, as discussed above) and then to use that as a bases to suggest those are the causes of the root-less and leaf-less anatomy. In fact, data in Figure 5 seems to argue against this, to some extent. The figure presents data on genes that show positive selection in the epiphytic Whisk Fern P. nudum. However, if the positively selected 74 of 803 genes (as stated for P. nudum) are leading causes for the epiphytic growth in P. nudum, then it would also predict they would be selected for in other epiphytic ferns, including O. pendulum. Based on the provided, that would not seem to be the case. Now, it might in fact be the case that the identified TF/TR genes and there evolution (loss and gain) are indeed directly involved the epiphytic growth and root-less/leaf-less anatomy of Whisk Ferns, but since TF/TR can vary in very subtle ways or can be used in a combinatorial way with other TF to impact transcriptional regulation, using the GO predictions alone is not sufficiently robust to have confidence these changes are the causative reason for the change in anatomy in the Whisk Fern. As I see it, the wording in the conclusions needs to reflect the uncertainty and speculator nature of these conclusions from these data.
Response 5: Thank you for the constructive advice. Firstly, we also recongaized that RNAseq has some problems in predicted genes/encoded proteins for TF/TR. Our views are as follows: Under the background of few genome-wide data of whisk ferns, nobody knows what happened on the genetic or structural information of whisk ferns during the evolution process. Using RNAseq to insight into the adaptive evolution of ferns on TF/TR from the prespectives of phylotranscriptomics and comparative transcriptomics remains an effective strategy, such as Qi et al., (2018) and Zhang et al., (2019). Secondly, in this study, we performed a reconstruction of TF/TR to identify TF/TR gene families that had changed corresponding to morphological development in whisk ferns and ophioglossoid ferns. Our results showed that the family sizes of some TF/TR related to morphological development (e.g., the GRF family and the MADS-MIKC family) were reduced in epiphytic whisk ferns (P. nudum and T. tannensis) and epiphytic O. pendulum (Table 1). This result is consistent with your suggestion. Thirdly, GO annoation was manily used to assess gene functions in this study, reflecting the reliability of adaptive evolution results. We also strongly agree that the use of molecular biology methods to explore the interaction between TF/TR is an effective way, but this is beyond the current conditions of our laboretory. We have seriously considered this suggestion, which is helpful in our next works.
Suggested specific edits and/or questions, ordered chronologically by Sections, Paragraphs, and Lines (“L”) in manuscript:
Point 6: Lines 57-61: Related to classification and evolution of ferns, the Smith et al. “A Classification Extant Ferns” (2006, Taxon. Vol 55, pp 705-731) is often cited.
Response 6: The Smith et al., 2006 (A Classification Extant Ferns) and the PPGI classification system proposed by Smith et al., 2017 both are frequently cited. Thanks for the reminder, we have cited the reference published by Smith et al., in 2017.
Point 7: Line 78 (and other locations in manuscript including figures): Some sources now use the name Physcomitrium patens instead of Physcomitrella patens name used in the manuscript. This includes the Phytozome database of the genetic sequence data that the authors cite.
Response 7: We have carefully checked Latin name and revised it according to the latest plant list.
Point 8: Lines 93-95: Data include both Transcriptomes and Proteomes (since used predicted protein sequences). Further, there needs some general information about RNAseq method with regard to age of plants, how grown, and what tissues/organs used. Full details need to be in the Materials and Methods (Lines 274-282), but some general information would be good to include in the Results.
Response 8: Great advice. We have re-described this part of Materials and Methods and Results, respectively, adding information about the source of sample, what tissues used, assessment of transcriptomes, and the other related.
Point 9: Line 97: Need to make it clear the protein sequences are “predicted hypothetical” sequences from the transcriptomes and not from direct protein sequencing.
Response 9: Thanks for the reminder, we have described in the method that the protein sequence are predicted by software TransDecoder.
Point 10: Line 99: the wording, “…from the 11 genomic data were …” is awkward and should be changed.
Response 10: Great advice. we have removed and re-descried this part as follows: A total of 348,947 non-redundant protein sequences were sorted into 30,051 orthogroups, which covered 82.3% of genes in all non-redundant sequences.
Point 11: Figure 2: The font color used (especially yellow font) is nearly impossible to read. Change font size and/or color to be readable.
Response 11: ok, we have changed the font size and color to make it more readable.
Point 12: Line 129: Cites Supplemental Table S2. In that table, the authors have column heads related to predicted TF/TR genes that are not clearly defined. They need to articulate the difference between what “Expanded TF/TR” means compared to “Genes gained”. Explain the difference in these data/gene numbers. Similarly for the “Contracted TF/TF” and “Genes Lost” headings.
Response 12: Supplemental Table S2 lists the exact number of TF/TR genes for each species, not the number of expansions or contractions.
Point 13: Lines 132 and 134: Mentions TF/TR “genes”, but I believe these really refer “gene families”. Please check and clarify, if necessary.
Response 13: Great advice, I have done, which seems more appropriate.
Point 14: Lines 133-136: This relates to my general concerns above and related to my “Third” general concern and that RNAseq data (from RNA) might “miss” genes if that gene or group of genes were not expressed or at sufficiently low level under particular growth conditions, developmental stage, or tissues used that they don’t appear in the RNA data. This might give a false predict of gene lost or “contraction”.
Response 14: This is a relatively professional advice. Nevertheless, whisk ferns, a non-model plant with a very large genome size (71G ~150G), faces several challenges for complete genome sequencing, including the complexity of complete genomes, the sequencing costs, and the computational resources required. In addiation, it has been shown that RNAseq can still provide useful insights in adaptive evolution analysis (Qi et al., 2018).
Point 15: Line 137: Cites Table 1 and claims of rapid evolution of TF/TR gene families. I already pointed out about what seems to be inconsistency in summary data in Table 1 versus full detailed data provided in Supplemental S2 Table. These need to be checked and resolved and/or explained better so that all data are consistent, and clear to readers.
Response 15: Based on the constructed species tree, a computational analysis of TF/TR family sizes was conducted (Supplementary Table S2) to study the expansion and contraction of TF/TR gene families involved in morphological development during the evolution of whisk ferns and their related species (Figure 3 and Supplementary Table S3). To explore the most significantly changed TF/TR involved in morphological develop-ment, TF/TR genes families with a rapid evolution were selected and are listed in Table 1.
Point 16: L139-140: claims that for GRF gene family there is, “significant contraction in both P. nudum and T. tannensis.” Table 1 states this is a loss of 5 genes (-5) for P. nudum and loss of 3 genes (-3) for T. tannensis. It is not clear what species these are compared to to know if there is a loss or not. And, even with that claim, as mentioned above, data in Supplemental S2 Table does not seem to support that claim. Please clarify.
Response 16: The significant changes (expansions or contractions) in the data in the summary Table 1 are relative to the most recent common ancestor, supported by the results of CAFÉ (Software for Computational Analysis of gene Family Evolution) analysis. While supplemental S2 lists the number of TF/TR based on the acquired transcriptomic/genomic data of each species, and the results are supported by the iTAK analysis.
Point 17: Figure 3: Not completely clear where the data for the provided pie charts that show gene families that appear to stay Constant, Expand, or Contract comes from. For example, the pie chart shown for the first node of the Psilotum nudum and Tmesipteris tannensis seems to match Tmesipteris tannensis exactly, when if it for the node it would seem it should be a mix or average that factors in both Psilotum nudum and Tmesipteris tannensis.
Response 17: The pie charts is the output of CAFÉ. We have marked the specific data on each node to make the pie charts more readable.
Point 18: Figure 3 Legend: I would indicate that text for species and data that is highlighted is “highlighted in red font.”.
Response 18: Thanks for your suggestion, we have made changes to the font size and color to make the figures more readable.
Point 19: Table 1: I have already pointed out what to me seems to be inconsistency in the summary data provided in Table 1 compared to the full data set provided in Supplementary S2 Table.
Response 19: The significant changes (expansions or contractions) in the data in the summary Table 1 are relative to the most recent common ancestor, supported by the results of CAFÉ (Software for Computational Analysis of gene Family Evolution) analysis. While supplemental S2 lists the number of TF/TR based on the acquired transcriptomic/genomic data of each species, and the results are supported by the iTAK analysis.
Point 20: Line 155: Awkward wording for, “…ferns carry usefully genetic information, …” Change to improve wording.
Response 20: Thanks for you suggestion, we re-describe as follows: Whisk ferns are characterized by rootless and leaf-like appendages, and applying phylogenetic strategies, especially from sequences of gene families involved in root in-itiation and leaf development, will help us understand the evolutionary history of root and leaf.
Point 21: Lines 159-162: This mentions different predicted subfamilies of WOX’s, including the AC-WOX, WC-WOX, and IC-WOX. These data come, in part, from the Supplementary S2 Table, but in that supplementary table it is only described as the HB-WOX gene family. If the AC-WOX, WC-WOX, and IC-WOX groups have been used before, please cite a reference for this. If not, then please explain the bases for how these names and implications of evolutionary timing was generated.
Response 21: The AC-WOX, WC-WOX, and IC-WOX groups have been used before, we have cited reference [5] and [35].
Point 22: Figure 4: The text of fern / plant species show in both A and B are nearly impossible to read, due to both small font size and low quality/resolution. Figure would need to be redone to make text readable.
Response 22: Thanks for your suggestion, we have made changes to the font size and color to make the figures more readable.
Point 23: Lines 182 and 186: Cited Supplementary Table S4 and Table S5. Note, these tables both have a type in the heading on upper right side. It should be “Annotation in EggNOG”.
Response 23: Thanks for your suggestion, we have revised.
Point 24: Lines 186-189: Claims functions related to positively selected genes in the Whisk Ferns, which as mentioned above, is based on RNAseq and protein predictions along with very general grouping into different Gene Ontology (GO) groups, shows that “the majority of these genes were involved in cellular responses to water and nutrient deficiency caused by the epiphytic environment, secondary wall biosynthesis, ….” This is way to speculative and here are no data provided or even cited related to the physiological claims made, let along that a list of genes based on hypothetical GO is what is causing this. This sentence needs to be either removed more written so it is clear that this is speculation based on very general data.
Response 24: GO annoation was manily used to assess gene functions in this study, reflecting the reliability of adaptive evolution results. We also strongly agree that the use of molecular biology methods to explore the interaction between TF/TR is an effective way, but this is beyond the current conditions of our laboretory. We have seriously considered this suggestion, which is helpful in our next works.
Point 25: Discussion: In general, some sections are very speculative and as worded implies a stronger conclusion than the data support, similar to point made above for Lines 186-189. Line 206-210: GO categorizing of new genes alone does not prove biochemical / physiological function. And, for some categories, such as “Kinase binding” there are so many genes/proteins that fit in this with vastly different biological functions, that it is very speculative.
Response 25: Dear reviewer, we admit our results lack wet experimental validation, but ferns are a special group, lacking of genome data and genetic transformation system, which lead to severe limitations on molecular validation. This research is manily discussed from a macro-perspective, which will lay a foundation for the next micro-research. As the reviewer1 said:”I found only a few reports on the molecular analysis of this species in the available literature”. All in all, your suggestion is very effective and we are willing to focus on it in the next work.
Point 26: Lines 232-236: Discussion about MADS-box subfamilies such as ANR1, AGL12, etc… in rice and Arabidopsis is fine, but was not made clear that these forms exist in Whisk Fern (nor were these mentioned in the Result). Further, these MADS-box proteins might have completely different functions in Whisk Fern compared to angiosperms. There was not a clear link made to Whisk Ferns, aside from MADS-box genes being present in these ferns, which is true for ALL eukaryotes (ferns to fish, plants to humans).
Response 26: We understand your point very well. However, most of ferns, expeciaaly whisk ferns, have very large genomes, and there is no effective reference database, which greatly limits what we can do. Now almost all of ferns research is based on the reference angiosperm database, expecially Arabidopsis thaliana. Of course, this is not an excuse to reject your suggestions, we just want you to understand our difficulties.

Round 2
Reviewer 2 Report
Plants Manuscript #: plants- 1651511
Authors: Z. Xia et al.
Title: Analysis of comparative transcriptome and positively selected genes reveal adaptive evolution in leaf-less and root-less whisk ferns.
This is a re-review of this manuscript after the authors re-submitted this work. The authors did respond to and addressed a number of my concerns, which is good. Unfortunately, however, there are number of my concerns, some very fundamental, that no response or explanation was given in either the “Authors’ Reply” not were changes/additions made in the updated manuscript.
I completely understand the issue of challenges when working with new groups of organisms for which very limited or no existing genomic data are available, and the importance of getting started by publishing a first “toe hold” on the data. However, there still needs to be complete description of the data provided in Methods and an accurate reflection in the text of the limits of the data and conclusions. I am concerned is that this “level” of completeness in the methods and clarity and accuracy in conclusions from the data have not yet been met for this manuscript. I list my on-going concerns below.
1) Authors said in “Author’s Reply” that they included the “Smith et al, 2017” citation. I do not see it in the update manuscript. It is neither shown in highlighted red text in updated manuscript nor did I find it when I used a “search” function in the updated manuscript PDF. Not sure of explanation for this?
2) My original concern was “Lines 93-95: Data include both Transcriptomes and Proteomes (since used predicted protein sequences). Further, there needs some general information about RNAseq method with regard to age of plants, how grown, and what tissues/organs used. Full details need to be in the Materials and Methods (Lines 274-282), but some general information would be good to include in the Results.”
Authors included some important additional information. But, it is not sufficient for a complete methods for the RNAseq data, especially since this is the first time doing, so setting precedent for method in Wisk Ferns.
What is still needed:
- For “age” of plants the authors just state “young”. Please provide how old in days/weeks.
- As for what organs/tissues were used for RNAseq, the authors state “young tissues”. This is way to vague. Is this the whole plant? Please add specific information.
- Still not information about how plants were grown (greenhouse? In the wild? Light conditions?). The only information provided is that some were, “obtained from China National Orchid Conservation Center”. Please provide sufficient information that is commonly provided for methods growing plants.
- There is still no information provided about number of replications for RNAseq method. This needs to be made clear. This is the method they use to generate all the sequence data and make all the conclusions from, but description of this method is still incomplete. If just one replicate, then there is a genuine concern that the data could be incomplete or missing for technical reasons.
3) My original concern was “Table 1: I have already pointed out what to me seems to be inconsistency in the summary data provided in Table 1 compared to the full data set provided in Supplementary S2 Table.”
The authors gave a reasonable response and explanation in the “Author’s Reply” about the comparison being to the most recent common ancestor. However, they need to add that explanation to the manuscript text itself, so readers would know this. This was not done.
4) The Typo (“Annoation…” which should be “Annotation…”) in the column heading in both Table S4 and Table S5 were still not corrected, although the authors stated in their reply, “Thanks for your suggestion, we have revised.” Again, concerned that the authors indicated they made the change, but the change does not show up on the updated documents.
5) At several points the conclusions in the Discussion were too speculative based on the current data. I pointed these out in detail in my first review, citing specific topic and line numbers for each case. This was mostly focused on the general grouping of genes and gene function predictions and specific physiological roles based on GO. This is a fine method in general, but it has limits when one tries to define very specific functions and then infer evolution based on these specific predictions. For example, Line 206-210: GO defined “Kinase binding” for some genes. Since Kinases Binding is very general and true for so many genes/proteins with widely ranging physiological impacts aside from “regulation” that GO category to infer somewhat specific evolutionary function on organ development is a very speculative.
The authors’ reply to these concerns were, in general, that studies of Whisk Ferns are in an early stage, which is fine. They also acknowledged that this approach has limitations. They go on and mention several times that these suggestions and concerns will be helpful “in their next work”. That is fine, but conclusions in this work still need to be based on solid data. I can accept some level of speculation, but the authors need to be more clear in the text of the manuscript with regard to the limitations on conclusions from these data. What I am asking the authors to do, at a minimum, is to include a statement in the Discussion acknowledging the limits to these conclusions and a statement that accurately factors in the uncertainty.
Currently, the authors have made no changes or additions in the Discussion to acknowledge that these data have limitations as to the conclusions.
Author Response
Response to Reviewer 2 Comments
Dear reviewer 2:
On behalf of my co-authors, we thanks again for giving us an opportunity to revise our manuscript, we appreciate editor and reviewers very much for your positive and constructive comments on our manuscript entitled “Analysis of comparative transcriptome and positively selected genes reveal adaptive evolution in leaf-less and root-less whisk ferns” (Manuscript ID: plants-1651511). We have studied your general and specific comments carefully and have tried our best to revise our manuscript according to the comments. The main corrections in the paper and responds are as flowing:
Point 1: Authors said in “Author’s Reply” that they included the “Smith et al, 2017” citation. I do not see it in the update manuscript. It is neither shown in highlighted red text in updated manuscript nor did I find it when I used a “search” function in the updated manuscript PDF. Not sure of explanation for this?
Response 1: Thanks again for the reminder. Is is not “Smith et al., 2017” but “Smith et al., 2016”. However, this is not the work of Smith alone, but the work organized by Eric Schuettpelz, Harald Schneider, Alan R. Smith et al., and the original research has recommended citing by the title of “The Pteridophyte Phylogeny Group” or “PPGI (2016)”. Sorry for not detailing the situation, this key reference has been cited in our manuscript in form “PPGI (2016)“.
Point 2: My original concern was “Lines 93-95: Data include both Transcriptomes and Proteomes (since used predicted protein sequences). Further, there needs some general information about RNAseq method with regard to age of plants, how grown, and what tissues/organs used. Full details need to be in the Materials and Methods (Lines 274-282), but some general information would be good to include in the Results.” Authors included some important additional information. But, it is not sufficient for a complete methods for the RNAseq data, especially since this is the first time doing, so setting precedent for method in Wisk Ferns.
Response 2: Thanks again for the reminder. We have re-described this part as follows: The other three field-collected adult species including terrestrial D. chinensis, epiphytic O. pendulum and terrestrial O. vulgatum were newly sequenced in this study. For transcriptome sequencing, the total plant transcriptomic RNA was isolated from mixed fresh tissues of leaf, stem and rhizome using the Tiangen Polysaccharide&Polyphenolics-rich RNAprep Pure Plant Kit.
Point 3: For “age” of plants the authors just state “young”. Please provide how old in days/weeks.
Response 3: Dear reviewer, the species we sequenced in this study were collected in field, and there are certain limitations in providing specific plant age, so we used “adult species” instead, which is more apprppriate when the samples is collected form fields.
Point 4: As for what organs/tissues were used for RNAseq, the authors state “young tissues”. This is way to vague. Is this the whole plant? Please add specific information.
Response 4: Thank you for your suggestion. As in response 2, we have re-descripted this part. As follows: the total plant transcriptomic RNA was isolated from mixed fresh tissues of leaf, stem and rhizome using the Tiangen Polysaccharide&Polyphenolics-rich RNAprep Pure Plant Kit.
Point 5: Still not information about how plants were grown (greenhouse? In the wild? Light conditions?). The only information provided is that some were, “obtained from China National Orchid Conservation Center”. Please provide sufficient information that is commonly provided for methods growing plants.
Response 5: Thank you for your suggestion. As in response 2, we have re-descripted this part. As follows: The other three field-collected adult species including terrestrial D. chinensis, epiphytic O. pendulum and terrestrial O. vulgatum were newly sequenced in this study.
Point 6: There is still no information provided about number of replications for RNAseq method. This needs to be made clear. This is the method they use to generate all the sequence data and make all the conclusions from, but description of this method is still incomplete. If just one replicate, then there is a genuine concern that the data could be incomplete or missing for technical reasons.
Response 6: Dear reviewer, I have to sincerely tell you that we only hav one data record per sample. In this study, the analysis of positively selected genes and comparative transcriptome was based on the backbone of phylogenetic result. To construct the phylogenetic tree, phylogenetic analyses were performed using single-copy orthologs for each species. Up to the present, unlike molecular analysis related to gene expression, most phylogenetic and comparative transcriptomic studies have one data per species. your concerns warrants some further thoughts. We must admit that the experimrntal data planning in the early stage of this study is indeed insufficient.
Point 7: My original concern was “Table 1: I have already pointed out what to me seems to be inconsistency in the summary data provided in Table 1 compared to the full data set provided in Supplementary S2 Table.” The authors gave a reasonable response and explanation in the “Author’s Reply” about the comparison being to the most recent common ancestor. However, they need to add that explanation to the manuscript text itself, so readers would know this. This was not done.
Response 7: Thanks again for the reminder, we are very sorry about responding to you but not adding it to the manuscript. We have added the following description to the Materials and Methods section: Additionally, the iTAK (v.18.12) program [15] was used for prediction and classification of plant TFs and TRs from all the protein sequences. Based on the constructed evolutionary tree, expansion and contraction of TFs and TRs relative to the most recent common ancestor were analyzed using the CAFÉ (v.4.2.1) program [48].
Point 8: The Typo (“Annoation…” which should be “Annotation…”) in the column heading in both Table S4 and Table S5 were still not corrected, although the authors stated in their reply, “Thanks for your suggestion, we have revised.” Again, concerned that the authors indicated they made the change, but the change does not show up on the updated documents.
Response 8: Thank you very much for the reminder. This spelling error has been corrected.
Point 8: At several points the conclusions in the Discussion were too speculative based on the current data. I pointed these out in detail in my first review, citing specific topic and line numbers for each case. This was mostly focused on the general grouping of genes and gene function predictions and specific physiological roles based on GO. This is a fine method in general, but it has limits when one tries to define very specific functions and then infer evolution based on these specific predictions. For example, Line 206-210: GO defined “Kinase binding” for some genes. Since Kinases Binding is very general and true for so many genes/proteins with widely ranging physiological impacts aside from “regulation” that GO category to infer somewhat specific evolutionary function on organ development is a very speculative.
The authors’ reply to these concerns were, in general, that studies of Whisk Ferns are in an early stage, which is fine. They also acknowledged that this approach has limitations. They go on and mention several times that these suggestions and concerns will be helpful “in their next work”. That is fine, but conclusions in this work still need to be based on solid data. I can accept some level of speculation, but the authors need to be more clear in the text of the manuscript with regard to the limitations on conclusions from these data. What I am asking the authors to do, at a minimum, is to include a statement in the Discussion acknowledging the limits to these conclusions and a statement that accurately factors in the uncertainty.
Currently, the authors have made no changes or additions in the Discussion to acknowledge that these data have limitations as to the conclusions.
Response 8: Thank you for you detailed review, and we are very sorry for not giving an accurate resopnse in the first reply. Your concerns about the conclusions of this study are justified, and we agree with you and would like to make a statement in the Discussion. As follows: Naturally, It cannot be denied that there are still some limitations in this research, and further research and discussion are needed. For example, this paper only considers the expressed genes for the species whose transcriptomes were sequenced, but in real life, RNAseq data could very easily miss genes if the RNA for a given gene is not expressed under the growth conditions or in the tissues used for RNA analysis. In addition, although one replicate per sample is reasonable for phylogenetic analysis in this research, it could result in incomplete or missing data for comparative transcrip-tome analysis. Moreover, in the absence of other valid validations, there is still uncertainty about GO annotations of gene functions with reference to EggNOG database. Nevertheless, our findings still preliminarily illustrate adaptive evolutionary patterns of whisk ferns in leaf-less and root-less. Therefore, more research is still needed, and more genomic data needs to be collected to establish a trend given the uncertainties in the experiments.
